# A Nonlinear Magnetoelastic Energy Model and Its Application in Domain Wall Velocity Prediction

**DOI:** 10.3390/s22145371

**Published:** 2022-07-19

**Authors:** Li-Bo Wu, Yu-Feng Fan, Feng-Bo Sun, Kai Yao, Yue-Sheng Wang

**Affiliations:** 1Department of Mechanics, School of Civil Engineering, Beijing Jiaotong University, Beijing 100044, China; libo_wu@bjtu.edu.cn; 2China Construction Second Bureau Installation Engineering Co., Ltd., Beijing 100176, China; yufengfan2020@163.com; 3China Construction Second Engineering Bureau Co., Ltd., Beijing 100160, China; sunfengbo@cscec.com; 4Department of Mechanics, School of Mechanical Engineering, Tianjin University, Tianjin 300350, China

**Keywords:** nonlinear magnetoelastic energy, magneto-crystalline constants, domain wall velocity

## Abstract

In this letter, we propose a nonlinear Magnetoelastic Energy (ME) with a material parameter related to electron interactions. An attenuating term is contained in the formula of the proposed nonlinear ME, which can predict the variation in the anisotropic magneto-crystalline constants induced by external stress more accurately than the classical linear ME. The domain wall velocity under stress and magnetic field can be predicted accurately based on the nonlinear ME. The proposed nonlinear ME model is concise and easy to use. It is important in sensor analysis and production, magneto-acoustic coupling motivation, magnetoelastic excitation, etc.

## 1. Introduction

Magnetoelastic Energy (ME) is essential in the guidance of magneto-acoustic coupling motivation [1,2,3,4], sensor production [5,6,7,8], and magnetoelastic excitation [9,10,11]. Classical Magnetoelastic Energy (ME) is a linear stress function that needs to be improved when predicting some specific aspects. However, many experiments indicate that the ME exhibits nonlinearity with increasing stress. The classical linear ME density can be expressed as $E_{me}=-3/2\;\lambda_{s0}\sigma \cos\theta_{\sigma }$ [12], where $\lambda_{s0}$ is the saturation magnetostriction coefficient without stress, σ is the stress, and $\theta_{\sigma }$ is the angle between the stress and magnetization. The Hamiltonian of the linear ME under displacement field $u\left(r\right)$ is generally expressed as [2,13]:
(1)$$H_{\mathrm{me}}=\frac{1}{s^{2}}\sum \int_{V}^{\;}B_{\alpha \beta }s_{\alpha }\left(r\right)s_{\beta }\left(r\right)\varepsilon_{\alpha \beta }\left(r\right)dr,$$
where $r=\left(x,y,z\right)$, α, β=x, y, z; s is the saturation spin density; *V* is the volume; $\varepsilon_{\alpha \beta }\left(r\right)$ is the linear strain component which can be expressed as $\varepsilon_{\alpha \beta }\left(r\right)=\left[\partial u_{\beta }\left(r\right)/\partial r_{\alpha }+\partial u_{\alpha }\left(r\right)/\partial r_{\beta }\right]/2$; $B_{\alpha \beta }$ is the magnetoelastic anisotropic constant; Ref. [2] and the Einstein summation convention is assumed.

Magnetization results from the electron’s spin, which is related to the lattice parameters [1,12]. The magneto-elastic coupling effects are mainly relevant to the exchange field, spin-orbit coupling, etc. [12,13]. According to Refs. [14,15], the primary mechanism of the interactions between atoms (A and B) and electrons (a and b) are schematically plotted in Figure 1, where ***r***_Aa_, ***r***_Bb_, ***r***_AB_, and ***r***_ab_ are position vectors. The red diamonds and green circles are the impacts of electrons a and b. The electrons’ interactions are the primary influence factors of magnetization. It can be seen that the interactions of the electrons are related to the position vectors between electrons $r_{\mathrm{ab}}$. The expectation of electrons’ distance $\bar{\left|r_{\mathrm{ab}}\right|}$ is also the function of *Ψ.* In addition, *Ψ* is the function of the nucleus position vector $r_{\mathrm{AB}}$. Thus, $\bar{\left|r_{\mathrm{ab}}\right|}$ can be expressed as:(2)$$\bar{\left|r_{\mathrm{ab}}\right|}=\int \Psi_{A}^{*}\left(r_{\mathrm{AB}}\right)\Psi_{B}^{*}\left(r_{\mathrm{AB}}\right)\left|r_{\mathrm{ab}}\right|\Psi_{A}\left(r_{\mathrm{AB}}\right)\Psi_{B}\left(r_{\mathrm{AB}}\right)dr,$$
where * means taking the conjugate.

The above is mainly the primary mechanism of the magnetoelastic effect [2,13]. ME is nonlinearly related to the nucleus distance. However, the classical linear ME includes only linear terms of Taylor’s series [16,17,18]. The linear ME can describe the magnetoelastic behaviors under small deformation [1,2]. However, it becomes more ineffective in representing nonlinear behaviors with increasing deformation. Furthermore, if the higher-order nonlinear terms are taken to describe the nonlinear behaviors, the number of expansion coefficients to be determined increases rapidly, which is inconvenient to use. As far as we know, the constructive nonlinear ME is rarely reported, which is essential in the magnetoelastic behaviors under larger deformation. 

In this letter, we construct a nonlinear ME with the material parameter to better and more conveniently describe nonlinear magnetoelastic behaviors. Then, the validity of the model was verified, and the model was applied in the prediction of the domain wall velocity under stress and a magnetic field, which is important in sensor production.

## 2. Model Construction

The derivation of the ME’s density is mainly divided into three steps [16,17,18]. Firstly, the ME is expanded to the form of magneto-crystalline anisotropy energy in Taylor’s series, and the first-order terms are taken as the ME approximately. Secondly, the expansion coefficients are solved under the equilibrium status without stress. Finally, the ME is obtained under the stress field. 

In this letter, we construct a new function basis to expand ME by considering the following facts: (1) the new function basis should be complete and orthogonal; (2) the higher-order terms of ME should tend to be zero and be negligible; (3) material parameters should be included in the new function basis to describe different magnetoelasticity for various materials; (4) the increasing rate of ME is related to deformation [19,20]; and (5) ME increases more slowly with the increasing deformation [19,20]. 

Based on the above analysis, a series of new function bases with material parameters $\left[1,xe^{-\left|\vartheta x\right|},x^{2}e^{-{\left|\vartheta x\right|}^{2}},\ldots x^{n}e^{-{\left|\vartheta x\right|}^{n}}\right]$ are chosen instead of the polynomial function basis $\left[1,x,x^{2},\ldots x^{n}\right]$, where $x=\varepsilon_{ij}$ is the strain component that is generally less than 1, and ϑ is the nonlinear material parameter related to the electron interactions. $e^{-\left|\vartheta \varepsilon_{ij}\right|}$ plays slight attenuating effects under larger strain (e.g., $\varepsilon >10^{-2}$), and $\vartheta \varepsilon_{ij}$ is generally less than 10 by the nature of function $xe^{-\left|\vartheta x\right|}$. Thus, it is reasonable to assume that $\vartheta <10^{3}$ in general. In addition, it is noted that $x^{n}e^{-{\left|\vartheta x\right|}^{n}}$ is closer to 0 as fast as $x^{n}$. Thus, the magneto-crystalline anisotropy energy density can be expressed as [12,13]: (3)$$E_{k}=E_{k}^{0}+\underset{i\geq j}{\sum }\left(\frac{\partial E_{k}}{\partial \left(\varepsilon_{ij}e^{-\left|\vartheta \varepsilon_{ij}\right|}\right)}\right)\varepsilon_{ij}e^{-\left|\vartheta \varepsilon_{ij}\right|}+\ldots ,$$

The first term ($E_{k}^{0}$) on the right-hand side of the above equation is the magneto-crystalline anisotropy energy density without stress. The remaining terms are the ME density (denoted by $E_{me}$), which can be viewed as the variation in the magneto-crystalline anisotropy energy density under stress. Generally, $\varepsilon_{ij}\ll 1$, and therefore, the second and higher-order terms are neglected. The expansion coefficients $\partial E_{k}/\partial \left(\varepsilon_{ij}e^{-\left|\vartheta \varepsilon_{ij}\right|}\right)$ are related to the direction cosine $\left(\alpha_{1},\alpha_{2},\alpha_{3}\right)$ of the magnetization vector. For the cubic crystal symmetry, the following equations are reasonable [13,16,18]:(4)$$\frac{\partial E_{k}}{\partial \left(\varepsilon_{ii}e^{-\left|\vartheta \varepsilon_{ii}\right|}\right)}=B_{1}\alpha_{i}^{2},\;\;\;i=1,2,3,$$
(5)$$\frac{\partial E_{k}}{\partial \left(\varepsilon_{ij}e^{-\left|\vartheta \varepsilon_{ij}\right|}\right)}=B_{2}\alpha_{i}\alpha_{j},\;\;\;\;i,j=1,2,3\;\&\;i>j,$$
where $B_{1}$ and $B_{2}$ are the magnetoelastic coupling coefficients to be determined.

Therefore, the nonlinear ME density can be expressed as: (6)$$E_{me}=B_{1}\overset{1,2,3}{\underset{i}{\sum }}\alpha_{i}^{2}\varepsilon_{ii}e^{-\left|\vartheta \varepsilon_{ii}\right|}+B_{2}\overset{1,2,3}{\underset{i>j}{\sum }}\alpha_{i}\alpha_{j}\varepsilon_{ij}e^{-\left|\vartheta \varepsilon_{ij}\right|}.$$

$B_{1}$ and $B_{2}$ can be solved based on the equilibrium conditions without external stress. Here, the free energy density (*E*) in the ferromagnetic crystal includes magneto-crystalline anisotropy energy density, ME density, and elastic energy density [16,18]. Only the magnetostrictive strain (denoted by $\varepsilon_{ij}^{\lambda }$) exits in ferromagnetic materials when no external stress is applied [12,13]. Thus, *E* can be expressed as:$$E=K_{1}\left(\alpha_{1}^{2}\alpha_{2}^{2}+\alpha_{2}^{2}\alpha_{3}^{2}+\alpha_{3}^{2}\alpha_{1}^{2}\right)$$
$$\;\;\;\;\;\;+B_{1}\overset{1,2,3}{\underset{i}{\sum }}\alpha_{i}^{2}\varepsilon_{ii}^{\lambda }e^{-\left|\vartheta \varepsilon_{ii}^{\lambda }\right|}+B_{2}\overset{1,2,3}{\underset{i>j}{\sum }}\alpha_{i}\alpha_{j}\varepsilon_{ij}^{\lambda }e^{-\left|\vartheta \varepsilon_{ij}^{\lambda }\right|}$$
(7)$$\;\;\;\;\;\;+\frac{1}{2}c_{11}\overset{1,2,3}{\underset{i}{\sum }}{\left(\varepsilon_{ii}^{\lambda }\right)}^{2}+\frac{1}{2}c_{44}\overset{1,2,3}{\underset{i>j}{\sum }}{\left(\varepsilon_{ij}^{\lambda }\right)}^{2}+c_{12}\overset{1,2,3}{\underset{i>j}{\sum }}\varepsilon_{ii}^{\lambda }\varepsilon_{jj}^{\lambda },$$
where $c_{11}$, $c_{12}$, and $c_{44}$ are elastic constants, and $K_{1}$ is the magneto-crystalline anisotropy constant. The first term on the right-hand side of the above equation is the magneto-crystalline anisotropy energy density $E_{k}^{0}$. The sum of the last three items is the elastic energy density $E_{el}$ for a cubic crystal. Then, $B_{1}$ and $B_{2}$ can be solved based on the equilibrium conditions:(8)$$\frac{\partial E}{\partial \left(\varepsilon_{ii}\right)}=B_{1}\alpha_{i}^{2}e^{-\left|\vartheta \varepsilon_{ii}^{\lambda }\right|}\left(1-\left|\vartheta \varepsilon_{ii}^{\lambda }\right|\right)+c_{11}\varepsilon_{ii}^{\lambda }+c_{12}\left(\varepsilon_{jj}^{\lambda }+\varepsilon_{kk}^{\lambda }\right)=0,\;i,j,k=1,2,3\;\&\;k\neq j\neq i,\;$$
(9)$$\frac{\partial E}{\partial \left(\varepsilon_{ij}\right)}=B_{2}\alpha_{i}\alpha_{j}e^{-\left|\vartheta \varepsilon_{ij}^{\lambda }\right|}\left(1-\left|\vartheta \varepsilon_{ij}^{\lambda }\right|\right)+c_{44}\varepsilon_{ij}^{\lambda }=0,\;i,j=1,2,3\;\&\;i>j.\;$$

Equations (8) and (9) are complicated to solve directly. However, the magnetostriction of non-giant magnetostrictive material is generally about 10^−5^ [21] and $\vartheta <10^{3}$ based on the above analysis. Therefore, $\vartheta \varepsilon_{ij}^{\lambda }$ is near 0, and $e^{-\left|\vartheta \varepsilon_{ij}^{\lambda }\right|}≅1$. Thus, the term $B_{1}\alpha_{i}^{2}e^{-\left|\vartheta \varepsilon_{ij}^{\lambda }\right|}\left|\vartheta \varepsilon_{ij}^{\lambda }\right|$ in the above equations can be ignored. Then, Equations (8) and (9) can be simplified as:(10)$$B_{1}\alpha_{i}^{2}+c_{11}\varepsilon_{ii}^{\lambda }+c_{12}\left(\varepsilon_{jj}^{\lambda }+\varepsilon_{kk}^{\lambda }\right)=0,\;\;\;i,j,k=1,2,3\;\&\;k\neq j\neq i,$$
(11)$$B_{2}\alpha_{i}\alpha_{j}+c_{44}\varepsilon_{ij}^{\lambda }=0,\;\;\;\;i,j=1,2,3\;\&\;i>j.\;$$

The magnetostrictive strains $\varepsilon_{ii}^{\lambda }$ and $\varepsilon_{ij}^{\lambda }$ can be solved as:(12)$$\varepsilon_{ii}^{\lambda }=\frac{B_{1}\left[c_{12}-\alpha_{i}^{2}\left(c_{11}+2c_{12}\right)\right]}{\left(c_{11}-c_{12}\right)\left(c_{11}+2c_{12}\right)},\;i=1,2,3$$
(13)$$\varepsilon_{ij}^{\lambda }=-\frac{B_{2}\alpha_{i}\alpha_{j}}{c_{44}},\;i\neq j$$

The micro-statistical method [22] is applied to construct the relationship between the coefficients $B_{1/2\;}$ and saturation magnetostriction $\lambda_{s}$, which can be measured by experiments. The following equations can be obtained for cubic crystals:(14)$$\lambda_{s}\left[100\right]=-\frac{B_{1}}{\left(c_{11}-c_{12}\right)}\left(1-\int_{0}^{2\pi }d\varphi \int_{0}^{\pi }\frac{1}{4\pi }\cos^{2}\theta \sin\theta d\theta \right),{\;}^{\;}$$
(15)$$\lambda_{s}\left[111\right]=-\frac{B_{2}}{c_{44}}\left(\frac{1}{3}-\int_{0}^{2\pi }\frac{1}{4\pi }\sin\varphi \cos\varphi d\varphi \int_{0}^{\pi }\sin\theta d\theta \right),\;$$
where $\lambda_{s}\left[100\right]$ and $\lambda_{s}\left[111\right]$ are the saturation magnetostrictions without stress along [100] and [111], respectively. Then, $B_{1}$ and $B_{2}$ are solved as:(16)$$B_{1}=-\frac{3}{2}\lambda_{s}\left[100\right]\left(c_{11}-c_{12}\right),\;$$
(17)$$B_{2}=-3\lambda_{s}\left[111\right]c_{44}.\;$$

Considering a simple case, the external stress tensor can be expressed as $\sigma_{ij}=\sigma \gamma_{ij}$, where $\gamma_{ij}$ is the direction cosine of the stress. The stress energy density $E_{\sigma }=\underset{i\geq j}{\sum }\sigma_{ij}\varepsilon_{ij}$ should be added in the free energy density *E*. Thus, the equilibrium conditions, Equations (8) and (9), change to:(18)$$\frac{\partial E}{\partial \left(\varepsilon_{ii}\right)}=B_{1}\alpha_{i}^{2}e^{-\left|\vartheta \varepsilon_{ii}\right|}\left(1-\left|\vartheta \varepsilon_{ii}\right|\right)+c_{11}\varepsilon_{ii}+c_{12}\left(\varepsilon_{jj}+\varepsilon_{kk}\right)-\sigma \gamma_{i}^{2}=0,\;.\;\;i,j,k=1,2,3\;\&\;k\neq j\neq i,\;$$
(19)$$\frac{\partial E}{\partial \left(\varepsilon_{ij}\right)}=B_{2}\alpha_{i}\alpha_{j}e^{-\left|\vartheta \varepsilon_{ii}\right|}\left(1-\left|\vartheta \varepsilon_{ij}\right|\right)+c_{44}\varepsilon_{ij}-\sigma \gamma_{i}\gamma_{j}=0,\;\;\;i,j=1,2,3\;\&\;i>j.$$

Generally, the magnetostriction is less than 10^−5^. Thus, $B_{1}$ and $B_{2}$ are far less than elastic constants. $e^{-\left|\vartheta \varepsilon_{ij}\right|}$ is less than 1, and $\left|\vartheta \varepsilon_{ii}\right|e^{-\left|\vartheta \varepsilon_{ij}\right|}$ is less than $e^{-1}$. As discussed above, the first terms on the right-hand side are far less than the second terms in Equations (18) and (19). Therefore, the first terms on the right-hand side can be ignored. The strain components are solved as: (20)$$\varepsilon_{ii}=\frac{\sigma \left[c_{12}-\gamma_{i}^{2}\left(c_{11}+2c_{12}\right)\right]}{\left(c_{11}-c_{12}\right)\left(c_{11}+2c_{12}\right)},\;$$
(21)$$\varepsilon_{ij}=\frac{\sigma \gamma_{i}\gamma_{j}}{c_{44}},\;i\neq j.\;$$

With the substitution of the magnetoelastic coupling coefficients ($B_{1}$ and $B_{2}$) and the direction-dependent terms of strain components into Equation (6), the nonlinear ME density, $E_{me}$, is obtained as:(22)$$E_{me}=-\frac{3}{2}\lambda_{s}\left[100\right]\sigma \overset{1,2,3}{\underset{i}{\sum }}\alpha_{i}^{2}\gamma_{i}^{2}e^{-\left|\frac{\vartheta \sigma \gamma_{i}^{2}}{\left(c_{11}-c_{12}\right)}\right|}-3\lambda_{s}\left[111\right]\sigma \overset{1,2,3}{\underset{i>j}{\sum }}\alpha_{i}\alpha_{j}\gamma_{i}\gamma_{j}e^{-\left|\frac{\vartheta \sigma \gamma_{i}\gamma_{j}}{c_{44}}\right|}.\;$$

For an isotropic material, $\lambda_{s}\left[100\right]=\lambda_{s}\left[111\right]=\lambda_{s0}$, then Equation (22) can be written as:(23)$$E_{me}=\lambda_{s0}\sigma \left(-\frac{3}{2}\overset{1,2,3}{\underset{i}{\sum }}\alpha_{i}^{2}\gamma_{i}^{2}e^{-\left|\frac{\vartheta \sigma \gamma_{i}^{2}}{\left(c_{11}-c_{12}\right)}\right|}-3\overset{1,2,3}{\underset{i>j}{\sum }}\alpha_{i}\alpha_{j}\gamma_{i}\gamma_{j}e^{-\left|\frac{\vartheta \sigma \gamma_{i}\gamma_{j}}{c_{44}}\right|}\right).\;$$

Under the uniaxial stress ($\gamma_{i}=1$, $\gamma_{j}=\gamma_{k}=0$, i≠j≠k), Equation (23) can be written as:(24)$$E_{me}=-\frac{3}{2}\;\lambda_{s0}\sigma \cos\theta_{\sigma }e^{-\left|\frac{\vartheta \sigma }{\left(c_{11}-c_{12}\right)}\right|},$$
where $\theta_{\sigma }$ is the angle between the stress and the magnetization.

The Hamiltonian of the nonlinear ME under displacement field ***u***(***r***) can be applied in nanoscale fields. It can be expressed as:(25)$$H_{\mathrm{me}}=\frac{1}{s^{2}}\sum \int_{V}^{\;}B_{\alpha \beta }e^{-\left|\vartheta \varepsilon_{\alpha \beta }\left(r\right)\right|}s_{\alpha }\left(r\right)s_{\beta }\left(r\right)\varepsilon_{\alpha \beta }\left(r\right)dr.\;$$

## 3. Model Verification

The variations in the magneto-crystalline anisotropy constant of CoFeB induced by ME with the stress applied along the *x* and *y* directions are given in Figure 2 [19]. The measurement was taken in a uniaxial in-plane anisotropy of the CoFeB/PVDF system. The magneto-crystalline anisotropy energy can be expressed by $E_{k}^{0}=K_{U}\left(\alpha_{1}^{2}+\alpha_{2}^{2}\right)$, where $K_{U}$ is the magneto-crystalline anisotropy constant [23]. Considering both magnetization and stress along the *x* direction ($\alpha_{1}=\gamma_{1}=1$, $\alpha_{2}=\alpha_{3}=\gamma_{2}=\gamma_{3}=0$) or along the *y* direction ($\alpha_{2}=\gamma_{2}=1$, $\alpha_{1}=\alpha_{3}=\gamma_{1}=\gamma_{3}=0$), we have $E_{k}=E_{k}^{0}+E_{me}=K_{U}+E_{me}/\cos\theta_{\sigma }$. Then, $E_{me}/\cos\theta_{\sigma }$ can be considered as the variation of $K_{U}$ which is denoted as $∆K_{U}$, i.e., $∆K_{U}=E_{me}/\cos\theta_{\sigma }$. Figure 2a presents the angular dependence of the normalized remanent magnetization ($M_{r}/M_{s}$), where $M_{r}$ is the remanent magnetization, and $M_{s}$ is the saturation magnetization. It shows a uniaxial anisotropy, and the easy axis is along the *y* direction. It should be reasonable to assume that the values of ϑ are different in different directions when the distribution of magnetic particles varies according to the physical meaning of ϑ. Therefore, the values of ϑ for CoFeB along with *x* and *y* are taken as $\vartheta_{x}\;$= 45 and $\vartheta_{y}$ = 52, respectively. The film can be regarded as a two-dimensional material different from the three-dimensional material. Then, a reduction factor of one-half should be included approximately in the ME’s density [24,25,26,27]. The saturation magnetostriction of CoFeB along both the *x* and *y* directions is taken as $\lambda_{s0}=$31 ppm [20]. The relationship between the strain and stress of CoFeB is $\varepsilon_{x/y}=\sigma_{x/y}\left(1-\nu^{2}\right)/G$ [19,20], where *G* (~162 GPa [19,20]) is the elastic modulus, and ν (~0.3 [19,20]) is the Poisson’s ratio of CoFeB. Thus, $∆K_{U}$ calculated by the linear ME density and nonlinear ME density along the *x* and *y* directions, are given by:(26)$$∆\;K_{Ux/y}=-\frac{3}{4}\lambda_{s0}\sigma \;\;\;\;\left(\mathrm{Linear}\;\mathrm{ME}\right)$$
(27)$$∆\;K_{Ux/y}=-\frac{3}{4}\;\lambda_{s0}\sigma e^{-\left|\frac{\vartheta_{x/\;y}\sigma \left(1-\nu^{2}\right)}{G}\right|}\;\;\;\;\left(\mathrm{Nonlinear}\;\mathrm{ME}\right)\;$$

It is observed in Figure 2b,c that the measured $∆K_{u}$ along the *x* and *y* directions [19] (black lines with the square points) increases with the increasing stress. However, the rate of increase decreases, which is more obvious along the *y* direction than along the *x* direction. $∆K_{u}$ (blue lines with the triangular points) in Figure 2b,c predicted by the linear ME density exhibits linear growth along the *x* and *y* directions with the increasing stress. When the stress is small, the results predicted by the linear ME density are close to the experimental results in Ref. [19]. However, the predicted errors become larger with the increasing stress. In other words, the prediction for some specific aspects based on the linear ME needs to be improved. In addition, there was a problem predicting the anisotropy along the *x* and *y* directions based on the linear ME density. The $∆K_{u}$ predicted by the proposed nonlinear ME (red lines with the circular points) in Figure 2b,c exhibits nonlinear growth along the *x* and *y* directions. It can predict the anisotropy as well. The predicted errors by nonlinear ME remain small with the increasing stress.

The ME can be regarded as the variation in magneto-crystalline energy under stress [12,13]. The magneto-crystalline constant is the magneto-crystalline energy density, with the angle’s cosine being 1. It is demonstrated that the increasing trends of magneto-crystalline constants are nonlinear, see the experimental results in Figure 2b,c [19]. This phenomenon results from the interaction between magnetic particles that decay with the increasing distance between particles [12]. Compared with the linear ME density $E_{me}^{Linear}=-3/2\lambda_{s0}\sigma \cos\theta_{\sigma }$ [12], the proposed nonlinear ME density contains exponential terms and material parameters. It makes the nonlinear ME more capable of describing the decaying growth trend and the variations between materials of different magnetoelastic behaviors under larger deformation.

## 4. Model Application

The proposed nonlinear ME density can be used widely. According to the previous description, magnetic anisotropy is related to magneto-elastic energy. We predicted the effect of magnetic anisotropy induced by stress on the domain wall (DW) dynamics for Co-rich microwires based on the nonlinear ME density. The velocity of DW propagates along with the wire is known to be [28,29]:(28)$$v=S{\left(H-H_{0}\right)}^{\;}$$
where *H* is the axial magnetic field, *H*_0_ is the critical propagation field, and S is the DW mobility given by:(29)$$S=2\mu_{0}M_{s}/\beta$$
where β is the viscous damping coefficient [28,29]. Moreover, $\beta \approx M_{s}{\left[K/\left(A/a\right)\right]}^{1/2}$, where $M_{s}$ is the saturation magnetization, *A* is the exchange stiffness constant, a is the distance between magnetic particles, and $K=K_{0}+K_{me}$ is the magnetic ansitropy. Here, $K_{0}$ is the magnetic anisotropy without stress, and $K_{me}=-3/2\;\lambda_{s0}\sigma e^{-\left|\vartheta \sigma /G\right|}$ is the magnetic anisotropy induced by stress based on the proposed nonlinear ME.

As is known [28,29], the domain wall velocity is related to the interaction between magnetic particles, which decays with the increasing distance between particles. The viscous damping coefficient *β* decreases as the increasing stress within a certain range. The measured DW velocity on the magnetic field under stress is shown as the scatter points [28] in Figure 3. The calculated results based on the linear ME are shown as the lines in Figure 3a. It is shown that the domain wall velocity decreases with the increasing stress. It is obvious that *β* increases with the increasing stress. Then, *S* decreases, as can be seen from Equation (28). Thus, the calculated domain wall velocity based on the linear ME decreases with the increasing stress. It is different from the experimental results. The calculated results based on the proposed nonlinear ME are shown as the lines in Figure 3b. It is demonstrated that the DW velocity increases with the stress and the increasing magnetic field within the limited measurement range. The experimental and calculated results are in good agreement. The nonlinear magnetoelastic energy density can describe the nonlinear behaviors to a certain extent.

## 5. Concluding Remarks

In this letter, the nonlinear magnetoelastic energy is constructed by expanding magnetoelastic energy based on magneto-crystalline anisotropy energy by applying a new function basis with material parameters. It can describe the different materials’ nonlinear magnetoelastic behaviors. The coefficients are determined by saturation magnetostriction, which can be measured in experiments. The proposed nonlinear magnetoelastic energy can better predict the experimental results of the magneto-crystalline anisotropy constant variation and anisotropy under the stress field than the classical linear magnetoelastic energy. Based on the nonlinear magnetoelastic energy, the domain wall velocity under stress and the magnetic field can be predicted accurately. The Hamiltonian of the nonlinear ME applied in nanoscale fields is obtained. It has promising applications in a wider range of fields, e.g., sensor production, magneto-acoustic coupling motivation, magnetic memory method testing, magnetoelastic excitation, etc.

## Figures and Tables

**Figure 1 sensors-22-05371-f001:**
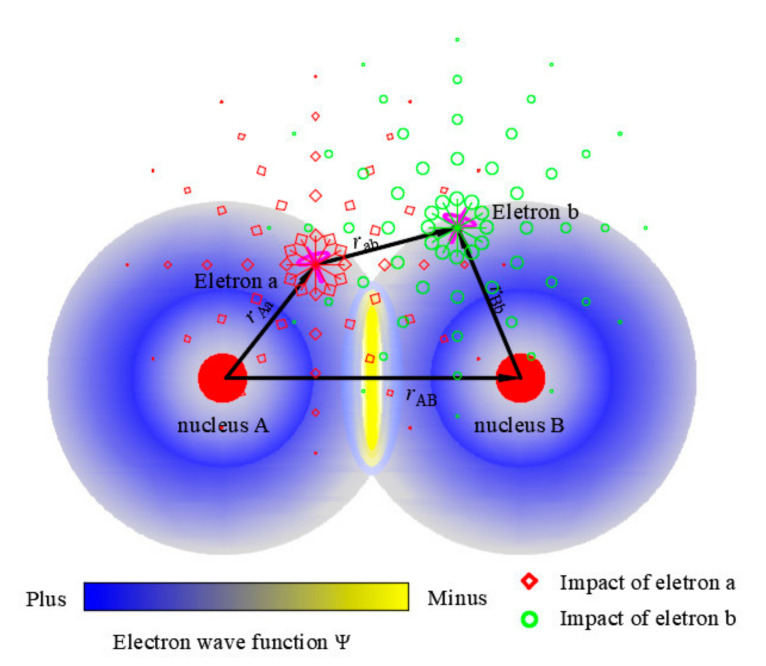
Primary mechanism of the interactions between atoms (A and B) and electrons (a and b).

**Figure 2 sensors-22-05371-f002:**
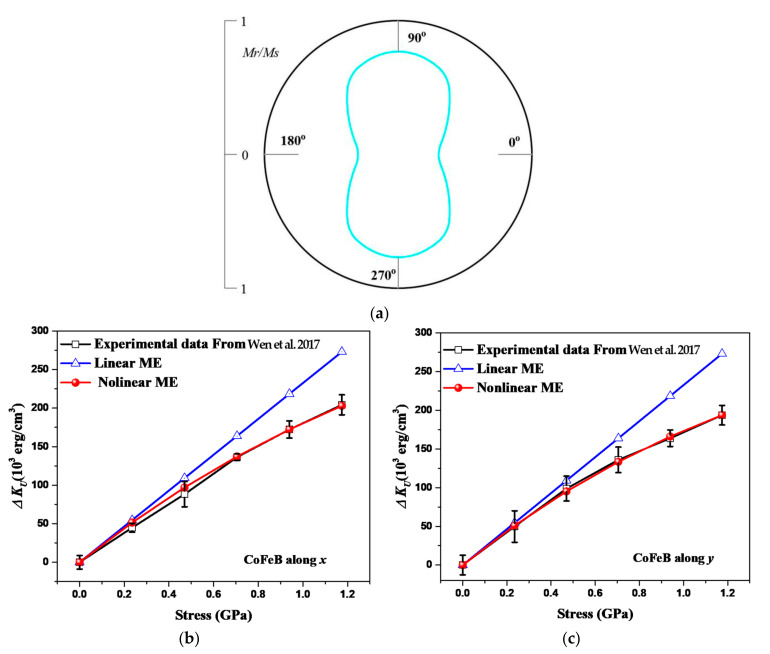
Comparison of the magnetic anisotropy constant variations in CoFeB by experiments [19] (Reproduced with permission from APPL. PHYS. LETT. 111(14), 142403 (2017). Copyright 2021 American Institute of Physics): (**a**) the definition of easy (*x*) and hard (*y*) magnetization directions, calculations by linear and proposed nonlinear ME (ME) along with *x* (**b**) and *y* (**c**) directions.

**Figure 3 sensors-22-05371-f003:**
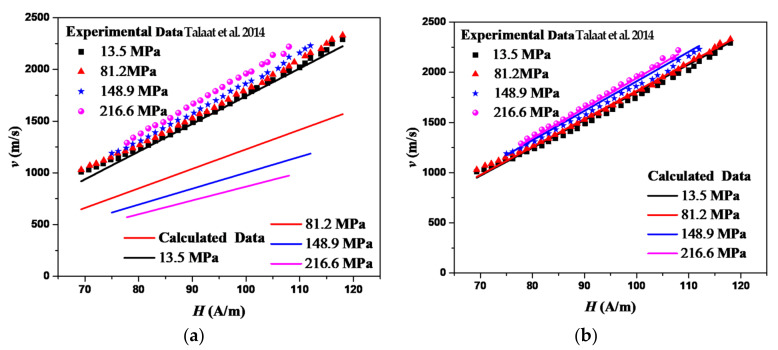
The predictions of DW velocity on magnetic field measured for Co_69.2_Fe_4.1_B_11.8_Si_13.8_C_1.1_ microwires [28] under different tensile stresses based on the linear ME. (Reproduced with permission from IEEE Transactions on Magnetics 50, 1–4 (2014). Copyright 2022 IEEE) (**a**) and the proposed nonlinear ME (**b**). The following parameters are used in the calculation: $K_{0}=8\;J/m^{3}$, A/a=1800 J/m, $\lambda_{s0}=1\times 10^{-7}$, $H_{0}=35\;A/m$, $\vartheta /G=3.1\times 10^{-8}{\;\mathrm{Pa}}^{-1}$, and *v* = 0.3.

## Data Availability

Not applicable.

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
