# Peer review of "A Nonlinear Magnetoelastic Energy Model and Its Application in Domain Wall Velocity Prediction"

_sensors, 2022, doi:10.3390/s22145371_

Round 1

Reviewer 1 Report

The authors propose a non-linear magnetoelastic energy with material parameters related to electron interactions. The proposed nonlinear ME model is also simple and easy to use. It may be useful in sensor analysis, production and magnetoelastic excitation. The article has a good theoretical depth. However, there are still some minor issues in the article and it is recommended to receive it after minor revisions.

1.      Figures 1 and 2(a) need to be redrawn, and at least the resolution should be increased to something like Figure 3.

2.      The "I" in the first word "In" in the abstract should not be in bold.

3.      The formula in the full text should be centered on it.

4.      Is the stress applied to the magnetostrictive material in the experimental section of the article applied directly to the material? Or was the stress applied to the carrier after the magnetostrictive material was attached to the carrier? Has the maximum stress that the material itself can withstand been considered?

Author Response

The authors propose a non-linear magnetoelastic energy with material parameters related to electron interactions. The proposed nonlinear ME model is also simple and easy to use. It may be useful in sensor analysis, production and magnetoelastic excitation. The article has a good theoretical depth. However, there are still some minor issues in the article and it is recommended to receive it after minor revisions.

Point 1: Figures 1 and 2(a) need to be redrawn, and at least the resolution should be increased to something like Figure 3.

Response 1: Thanks for your comments. We have redrawn Figures 1 and 2(a), see Page 2 and Page 6.

Point 2:The "I" in the first word "In" in the abstract should not be in bold.

Response 2: Thanks for your comments. We have corrected it, see Page 1.

Point 3:The formula in the full text should be centered on it.

Response 3: Thanks for your comments. We have centred the formula.

Point 4: Is the stress applied to the magnetostrictive material in the experimental section of the article applied directly to the material? Or was the stress applied to the carrier after the magnetostrictive material was attached to the carrier? Has the maximum stress that the material itself can withstand been considered?

Response 4: Thanks for your comments. The stress was applied to the carrier after the magnetostrictive material was attached to the carrier [Refs. 19, 28]. However there is no description for the material under the maximum stress in the Refs.

According to the Ref. 28: “Ta(2 nm)/CoFeB(40 nm)/Ta(4 nm) were deposited on 50 lm thick PVDF and Si substrates by dc magnetron sputtering at room temperature.”

As far as I'm concerned,the maximum stress value of the material is not reached within the measured range. Because there will be an obvious distortion in the magnetic domain wall propagation if the stress limit is reached.

Reviewer 2 Report

This paper presents a nonlinear magnetoelastic energy for domain wall velocity prediction, which is important for sensor production. A series of new function basis with material parameters is chosen instead of the polynomial function basis. It seems that the domain wall velocity under stress and magnetic field can be predicted accurately based on the nonlinear ME. There is one comment:

How to decide the the nonlinear material parameter related to the electron interactions.

Author Response

This paper presents a nonlinear magnetoelastic energy for domain wall velocity prediction, which is important for sensor production. A series of new function basis with material parameters is chosen instead of the polynomial function basis. It seems that the domain wall velocity under stress and magnetic field can be predicted accurately based on the nonlinear ME. There is one comment:

Point 1: How to decide the the nonlinear material parameter related to the electron interactions.

Response 1: Thanks for your comments. The CoFeBo is isotropic material. The nonlinear material parameter related to the electron interactions are empirical parameters. It is difficult to calculate it strictly at this stage. Thus, these values were taken after optimization here.

Reviewer 3 Report

In the paper, an  interesting nonlinear model of magnetoelastic energy is proposed. The approach described in the text can be useful for wide circle of researchers deal with different kinds of materials.
The reviewer has found only two small points to improve.
- The yellow diamonds in fig. 1 are almost non visible.
- Eqs. 7, 8, and 9 should be centred as other ones.

Author Response

In the paper, an interesting nonlinear model of magnetoelastic energy is proposed. The approach described in the text can be useful for wide circle of researchers deal with different kinds of materials.

The reviewer has found only two small points to improve.

-Point 1: The yellow diamonds in fig. 1 are almost non visible.

Response: Thanks for your comments. We have the yellow diamonds to red diamonds in fig. 1, see Page 2.

Point 2: Eqs. 7, 8, and 9 should be centred as other ones.

Response: Thanks for your comments. We have centred Eqs. 7, 8, and 9, see Page 3.

Reviewer 4 Report

This manuscript authored by Li-Bo Wu et al. proposes a nonlinear Magnetoelastic Energy(ME) model with the material parameter related to the electron interactions. This model can predict the variation of the anisotropic magneto-crystalline constants induced by external stress and predict the domain wall velocity under stress and magnetic field. The contents of this paper are interesting and meaningful. However, there are some concerns about the comparison between the experimental results and calculation data. I recommend the major revision before publication and the following points should be addressed.

1)For the manuscript title, there is a grammatical mistake, “A Nonlinear Magnetoelastic Energy Model” makes more sense.

2)For Figure 2, the magnetic anisotropy constant of CoFeB by experiments (black square lines) is different from the data in the original paper. (Fig. 6 in Wen, Xingcheng, et al. "Determination of stress-coefficient of magnetoelastic anisotropy in flexible amorphous CoFeB film by anisotropic magnetoresistance." Applied Physics Letters 111.14 (2017): 142403. ) Please explain this.

3)For Figure 3, authors miss the reproduction permission for the data.

4)Authors applied the nonlinear ME model in the domain wall velocity prediction and show the data in Figure 3, in which the nonlinear model shows better results than the linear model. However, in the original paper (Fig. 5, Talaat, Ahmed, et al. "Domain wall propagation in Co-based glass-coated microwires: Effect of stress annealing and tensile applied stresses." IEEE Transactions on Magnetics 50.11 (2014): 1-4.), Ahmed Talaat et al. presented the dependences of DW velocity on magnetic field measured in stress annealed Co69.2Fe4.1B11.8Si13.8C1.1 microwires at different annealing times, (a) 20 and (b) 45 min, under different tensile stresses. Please explain the reason for choosing data from Fig. 5(b), how about the calculation results for other situations.

Author Response

This manuscript authored by Li-Bo Wu et al. proposes a nonlinear Magnetoelastic Energy(ME) model with the material parameter related to the electron interactions. This model can predict the variation of the anisotropic magneto-crystalline constants induced by external stress and predict the domain wall velocity under stress and magnetic field. The contents of this paper are interesting and meaningful. However, there are some concerns about the comparison between the experimental results and calculation data. I recommend the major revision before publication and the following points should be addressed.

Point 1: For the manuscript title, there is a grammatical mistake, “A Nonlinear Magnetoelastic Energy Model” makes more sense.

Response 1: Thanks for your comments. We have changed the title to “A Nonlinear Magnetoelastic Energy Model”, see Page 1.

Point 2: For Figure 2, the magnetic anisotropy constant of CoFeB by experiments (black square lines) is different from the data in the original paper. (Fig. 6 in Wen, Xingcheng, et al. "Determination of stress-coefficient of magnetoelastic anisotropy in flexible amorphous CoFeB film by anisotropic magnetoresistance." Applied Physics Letters 111.14 (2017): 142403. ) Please explain this.

Response 2: Thanks for your comments. It is shown the variation value of the magnetic anisotropy constant ΔKU  in this paper. It is demonstrated the magnetic anisotropy constant KU in the original paper. And the initial value of the variation value and the value of the magnetic anisotropy constant is zero and nonzero, respectively.

Point 3: For Figure 3, authors miss the reproduction permission for the data.

Response 3: Thanks for your comments. We have added the reproduction permission for the data for Fig.3, see Page 7.

Point 4: Authors applied the nonlinear ME model in the domain wall velocity prediction and show the data in Figure 3, in which the nonlinear model shows better results than the linear model. However, in the original paper (Fig. 5, Talaat, Ahmed, et al. "Domain wall propagation in Co-based glass-coated microwires: Effect of stress annealing and tensile applied stresses." IEEE Transactions on Magnetics 50.11 (2014): 1-4.), Ahmed Talaat et al. presented the dependences of DW velocity on magnetic field measured in stress annealed Co69.2Fe4.1B11.8Si13.8C1.1 microwires at different annealing times, (a) 20 and (b) 45 min, under different tensile stresses. Please explain the reason for choosing data from Fig. 5(b), how about the calculation results for other situations.

Response 4: Thanks for your comments. The more sufficient the annealing time, the more sufficient the residual stress release. Then the more interference factor can be excluded. Thus, we presented the calculation results with the annealing times being 45 min which is more convincing. And there will be unquantifiable effects of residual stress on the experimental results for 20 min.Thus we have not calculated the results for 20 min annealing time

Round 2

Reviewer 4 Report

The authors have corrected the most points, however, for Point 1, I guess the authors misunderstood it. The title was suggested to be "A Nonlinear Magnetoelastic Energy Model and its Application in Domain Wall Velocity Prediction" rather than "A Nonlinear Magnetoelastic Energy Model". After fixing this, I suggest the acceptance of the manuscript. 

Author Response

Point 1: The authors have corrected the most points, however, for Point 1, I guess the authors misunderstood it. The title was suggested to be "A Nonlinear Magnetoelastic Energy Model and its Application in Domain Wall Velocity Prediction" rather than "A Nonlinear Magnetoelastic Energy Model". After fixing this, I suggest the acceptance of the manuscript.

Response 1: Thanks for your comments. We have changed the title to “A Nonlinear Magnetoelastic Energy Model and its Application in Domain Wall Velocity Prediction”, see Page 1.
